

# Offloading the computational complexity of transfer learning with generic features

Muhammad Safdar Ali Khan[1], Arif Husen[1,2], Shafaq Nisar[1], Hasnain Ahmed[1], Syed Shah Muhammad[1] and Shabib Aftab[1]

[1] Department of Computer Science and Information Technology, Virtual University of Pakistan, Lahore, Punjab, Pakistan

[2] Department of Computer Science, COMSATS Institute of Information Technology, Lahore, Punjab, Pakistan

## ABSTRACT

Deep learning approaches are generally complex, requiring extensive computational resources and having high time complexity. Transfer learning is a state-of-the-art approach to reducing the requirements of high computational resources by using pre-trained models without compromising accuracy and performance. In conventional studies, pre-trained models are trained on datasets from different but similar domains with many domain-specific features. The computational requirements of transfer learning are directly dependent on the number of features that include the domain-specific and the generic features. This article investigates the prospects of reducing the computational requirements of the transfer learning models by discarding domain-specific features from a pre-trained model. The approach is applied to breast cancer detection using the dataset curated breast imaging subset of the digital database for screening mammography and various performance metrics such as precision, accuracy, recall, F1-score, and computational requirements. It is seen that discarding the domain-specific features to a specific limit provides significant performance improvements as well as minimizes the computational requirements in terms of training time (reduced by approx. 12%), processor utilization (reduced approx. 25%), and memory usage (reduced approx. 22%). The proposed transfer learning strategy increases accuracy (approx. 7%) and offloads computational complexity expeditiously.

## INTRODUCTION

In recent years, deep learning (DL) applications have seen wide adoption. Several real-life improvements have been witnessed in different fields of life, such as image and voice detection (*Chen & Peng, 2023*), internet search, fraud identification (*Amiri et al., 2023*; *Nguyen & Reddi, 2023*), email/spam filtering (*Debnath & Kar, 2022*), and risk modeling in finance (*Liu & Yu, 2022*). It has been proven to provide new capabilities in different fields, such as Playing Go (*Gibney, 2016*), self-driven vehicles (*Ramos et al., 2017*), navigation, and object recognition (*Buduma & Locascio, 2017*). Moreover, DL models have outperformed other shallow learning techniques in various scientific fields and other areas (*Albrecht et al., 2017*; *Ge et al., 2020*; *Heidari et al., 2023*; *Kearnes et al., 2016*).

Corresponding author
Arif Husen, arifhrashid@gmail.com

Despite several benefits, DL models often suffer from the disadvantages of large amounts of memory, processing power, and storage to train and run (*Steiner et al., 2023*). The computational resources are expensive and time-consuming and may require specialized hardware, such as GPUs or TPUs, to achieve optimal performance (*Wang, Wei & Brooks, 2019*). Moreover, due to higher computational complexity, the training time and response time of the DL models are higher than those of other approaches (*Kearnes et al., 2016*; *Sapoval et al., 2022*). The transfer learning strategy has been used by various researchers, such as *Aslan et al. (2021)*, *Trappey, Trappey & Shih (2021)*, and *Ma et al. (2019)*, to address the above-stated issues.

Transfer learning not only relaxes the high computational requirements required to train DL models but also allows the application of the knowledge learned from different domains to a target domain, which is also similar but different (*Haykin, 2019*). Moreover, transfer learning is an efficiency-saving strategy in areas of ML that have high computational complexity image classification, detection, and natural language processing (*Paul, 2021*; *Ruder et al., 2019*; *Trappey, Trappey & Shih, 2021*).

This article focuses on improving classification accuracy and reducing the computational requirements of transfer learning. It analyses various state-of-the-art strategies and proposes a Generic Feature-Based TL (GFTL) strategy. The proposed strategy is applied to the Breast Cancer Detection and Classification (BCDC) problem to verify the computational requirements. The performance of the GFTL is evaluated in terms of accuracy, precision, memory and CPU utilization, and training time.

## Deep learning algorithms

DL algorithms are classified as a subset of machine learning techniques, which aim to replicate the complex architecture and functionality of the human brain to model and solve complex problems in computer vision. DL algorithms commonly use artificial neural networks (ANN) such as convolutional neural networks (CNN) and recurrent neural networks (RNN). According to IBM and Gartner, CNNs have a significant role in computer vision, enabling machines to identify objects and images with new accuracy levels (*IBM, 2023*). Many CNN models are available as part of the Keras library in Python (*Eli, Antiga & Viehmann, 2020*). The popular and latest CNN models are ResNet50 (*He et al., 2016*), MobileNet (*Howard et al., 2017*), AlexNet (*Krizhevsky, Sutskever & Hinton, 2017*), Inception V3 (*Guidang, 2019*), and LeNet (*Yanmei, Bo & Zhaomin, 2021*). The models have their strengths and certain advantages in image classification (*Du et al., 2023*). The standard CNN architectures pre-trained on well-known datasets are commonly used to accomplish transfer learning tasks (*Khalil et al., 2023a*). The CNN model has millions of trainable parameters and thousands of classes.

## Transfer learning strategies

Transfer learning is a DL strategy involving knowledge from one pre-trained model to another. The knowledge of the pre-trained model can be utilized to solve a new problem in a different domain (*Khamparia et al., 2021*). The primary idea behind transfer learning is to use what has been learned in one activity to enhance generalization in another

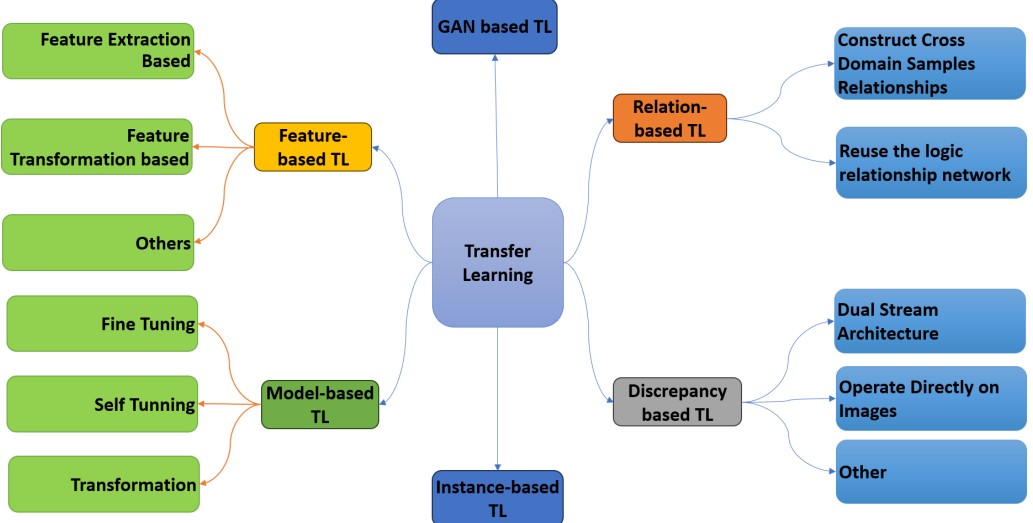

**Figure 1  Taxonomy of transfer learning.**

(*Saber et al., 2021*). Transfer learning has been widely used in computer vision to address the problem of scarce data in the medical imaging domain. Typically, fine-tuning is performed on the target data. However, fine-tuning increases the need for computational resources in transfer learning. Standard CNN models are used to accomplish transfer learning tasks. The models are multi-layer architectures pre-trained on ImageNet or some other well-known dataset. Pre-trained CNN models consist of two parts: the convolutional base and the classifier part. The convolutional base extracts different features from the data, while the classifier's job is to classify the data into different classes. There are three different ways to use these pre-trained CNN models, upon which transfer learning is categorized into six main categories, as shown in Fig. 1.

### GAN-based transfer learning

The application of generative adversarial networks (GANs) is often called GAN-based transfer learning in various studies conducted by *Elaraby, Barakat & Rezk (2022)*, *Srivastav, Bajpai & Srivastava (2021)* and *Yaqub et al. (2022)*. A GAN is a typical neural network structure used to generate distinct information samples exhibiting characteristics similar to an existing dataset. In GAN-based transfer learning (GANTL), a GAN plays a crucial role in facilitating the transfer of learned features from one domain to another. GAN models are trained with a source dataset containing a large amount of data on a particular input type, such as images, texts, or sounds. The primary aim of a GAN is to acquire knowledge regarding the underlying distribution of the given dataset and subsequently generate novel data instances that exhibit a high degree of resemblance to the original data.

### Relation-based transfer learning

The objective of relation-based transfer learning is to understand the relationships or similarities between seemingly unrelated tasks or domains (*Yang et al., 2020*). Learning

from one activity and applying it to another is the fundamental idea behind relation-based transfer learning. It accomplishes this by explicitly modeling the correlations or similarities between the two domains (*Legrand et al., 2021*). The two fundamental components of relational transfer learning are the source domain or task (from which information is being transferred) and the target domain or task (to which it is being transferred). Understanding the relation or mapping between the two domains or activities is the primary goal of relation-based transfer learning. This mapping focuses on the two domains' similarities, contrasts, and connections. Since relation-based transfer learning models the relationships between tasks or domains, it facilitates knowledge transfer more effectively. Recent studies have presented its several applications such as *Di, Shen & Chen (2019)*, *Nag et al. (2023)* and *Yu et al. (2018)*.

### Discrepancy-based transfer learning

The goal of discrepancy-based transfer learning (DiTL) is to simplify the learning process by lowering the discrepancies between the source and target domains (*Su et al., 2023*). This methodology aligns the distributions of the source and target datasets. After quantifying the discrepancies, alignment measures are implemented to minimize them. These strategies aim to enhance the similarity between the feature representations of the source and target data. It facilitates the effective transfer of knowledge acquired in the source domain to the target domain. DiTL has been used in computer vision in several recent studies, such as *Chen et al. (2022)*, *Nam et al. (2021)*, and *Yoon, Kang & Cho (2022)*.

### Instance-based transfer learning

An approach to transfer learning known as instance-based transfer learning focuses on modeling particular data instances, such as samples or examples, from a source domain to a target domain (*Wang, Huan & Zhu, 2019*). In contrast to traditional approaches to transfer learning that require the transfer of model parameters or features, instance-based transfer learning focuses on the transmission of individual data points or instances. The central idea revolves around choosing specific samples or instances from the source domain with the most significant relevance to the target domain. After that, the chosen instances from the source are transferred to the target activity or domain. It deals with a simple scenario with a lot of labeled data in the source domain but not much in the target domain (*Chowdhury, Annervaz & Dukkipati, 2019*; *Kim & Lee, 2022*; *Zhang et al., 2023*).

### Model-based transfer learning

Model-based transfer learning is a widely utilized technique in transfer learning. This strategy involves transferring knowledge from a pre-existing model, known as the source model, to a newly developed model called the target model (*Molina et al., 2021*). This technique aims to enhance the performance of the target model on a task or domain that is related but distinct from the original domain. Model-based transfer learning focuses on applying learned parameters from one model to another. A popular approach in this technique involves extracting features by deleting the output layer of the source model and utilizing the remaining layers. The parameters of the source model are used to initialize the target model, which is then fine-tuned using the target data.

*Feature-based transfer learning*

In feature-based transfer learning, the knowledge is transferred from an already-trained model (source model) to a new model (target model) with the help of the feature representations that the source model learned (*Chou, Wang & Lo, 2023*; *Molina et al., 2021*; *Shan et al., 2021*). Extracting and reusing pertinent features from the source model to boost the performance of the target model is the core emphasis of feature-based transfer learning. These methods modify the source features to generate a new representation of the features. This technique uses the features extracted from intermediate layer representations instead of the whole source model. These features are more generic and transferable across different fields since they extract basic information about the data. The extracted features are then fed into the target model as input features. Even when the target domain contains small, labeled data, feature-based transfer learning can improve performance by reusing valuable feature representations learned from big datasets, which saves time and resources (*Zhao, Shetty & Pan, 2017*).

## A review of breast cancer detection and classification

This article applies the GFTL to a critical domain that detects and classifies breast cancer. Cancer is a fatal disease that has spread rapidly worldwide over the last two decades. In the different types of cancers, breast cancer is recognized as the primary cause of a high death rate in women. *Ragab et al. (2021)* have studied breast cancer vulnerability across men and women. It has been concluded that although men are also vulnerable to breast cancer, it is a common disease in women. The ratio of breast cancer is increasing very rapidly all over the world, especially in Pakistan. Thus, particular focus is required for early and reliable detection methods of BCDC (*Jabeen et al., 2022*).

Breast cancer has several stages, based on which it is divided into different categories and sub-types. There are two significant categories of breast cancer: Invasive and non-invasive breast cancer. Non-invasive is not very dangerous, but the invasive type of breast cancer is the most dangerous because it spreads to other nearby organs (*Zahoor, Shoaib & Lali, 2022*). In the breast, the milk goes through an organization of minuscule tubes called ducts. Ducts are the most common spots where breast cancer attacks. Approximately 80% of all breast cancer cases are attributed to it, making it the most common type of breast cancer (*Khalil et al., 2023b*). In the sub-types of invasive breast cancer, invasive ductal carcinoma is the most dangerous and common kind of breast cancer that disturbs the milk ducts and nearby tissues (*Derakhshan & Reis-Filho, 2022*; *Diamantopoulou et al., 2022*).

Invasive lobular carcinoma is the other invasive type of breast cancer that affects lobes and their nearby tissues in the breast. Breast cancer could be benign (stage 1 tumor) or malignant (could be perilous). Benign cancers are not observed as unsafe because their cells are not very dangerous. Benign tissues grow slowly and do not extend to other body parts or affect nearby cells (*Mouabbi et al., 2022*). On the other hand, malignant cancers are more damaging and may extend beyond the other body parts from where they originated (*Ahmed et al., 2020*). Therefore, diagnosing and treating breast cancer earlier helps decrease the death rate (*Khan et al., 2019*).

### Common approaches for breast cancer diagnosis

Beginning in their age of forties or fifties, women at moderate risk of breast cancer should begin having medical check-ups and continue to do so. Which imaging technology to choose depends on several parameters, including breast density, risk factors, patient's age, and other characteristics. Imaging tests can be required more frequently or earlier in those with a family history of the disease or a gene that increases their risk of getting it (*Tan et al., 2023*).

Imaging technology has made finding and diagnosing breast cancer at the earliest stage easier. The patients can get help sooner and have better results. Different methods are used to detect BC: Magnetic resonance imaging (MRI) scan, histopathology, ultrasound scans, mammograms, and biopsy. Except for the last method, all other methods involve the analysis of the different images. A doctor can do the image analysis manually, or some automated systems can be used to analyze the image and diagnose cancer (*Durham et al., 2022*; *Han et al., 2022*).

When it comes to the diagnosis of breast cancer, according to the National Library of Medicine, mammography has been considered the gold standard (*Miller, 2001*). Mammography is an important screening test advised by pathologists at the initial stage of diagnosis. This method uses low-dose X-rays to obtain pictures of the breast and its tissue. BC detection using mammograms generated by X-ray machines is the best practice for pathologists who need to find malignant breast growth early (*Zahoor, Shoaib & Lali, 2022*). Mammograms can find abnormalities in the breast that a regular breast check-up might not be able to find. There are two standard mammogram views: the craniocaudal (CC) view and the medio lateral oblique (MLO), as shown in Fig. 2. In the CC view, the image is taken from the top of the breast downward; in the MLO, the image is taken from the side of the breast at an angle. The CC view is the common standard for routine screening and diagnostic mammography. The entire breast is shown when viewed from the top facing downward.

### Breast cancer detection and classification with DL techniques

DL has played a vital role in diagnosing different diseases. However, the efficient DL models are data-hungry, complex, and require extensive computational resources. For valuable results, DL requires training on thousands or even millions of images (*Houssein, Emam & Ali, 2022*). Such a considerable amount of data is unavailable in most medical imaging modalities. Training of a deep learning model on small data leads to overfitting problems. Deep learning models can overfit when they become excessively good at the tasks they are given to perform. Models that perform well on training data but struggle when applied to new patient data can produce incorrect diagnostic results. Concerning time and resources, it has been concluded that training is costly in DL.

Recently, transfer learning has proven to be an effective technique for detecting and classifying breast cancer in medical images. Medical experts can exploit it to develop diagnostic tools that are more accurate and take less time. It can lead to earlier detection of BC and better care for patients. Concerns about privacy and lack of data can make it hard to get a large and varied dataset in medical imaging (mammograms) for training deep

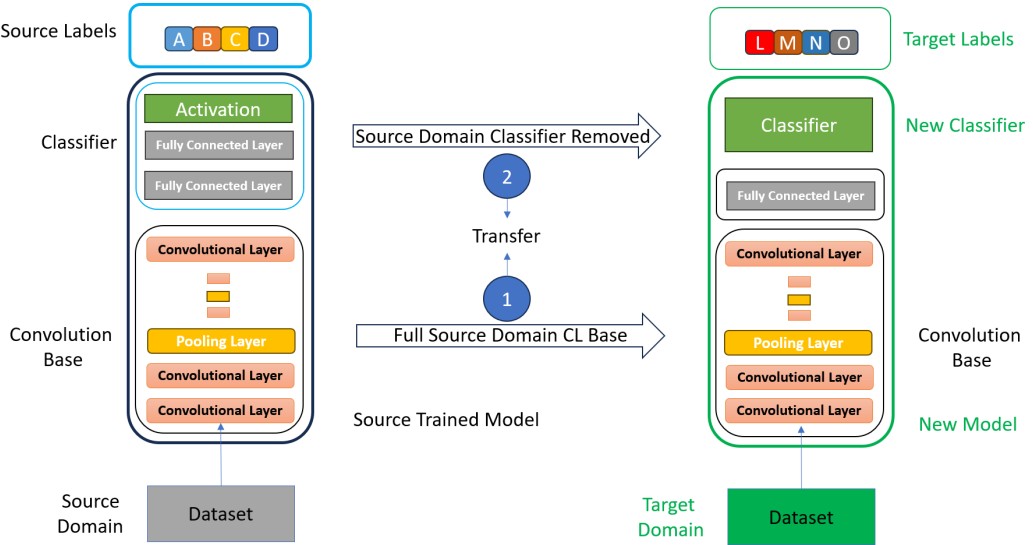

**Figure 2** Conventional transfer learning approach, adopted from *Saber et al. (2021)*.

learning models. Transfer learning helps avoid these problems by utilizing the pre-trained models already trained on large but unrelated general datasets like ImageNet. It can bridge several gaps, such as the differences between the available data from medical imaging and the needs of DL models.

ImageNet is a widely used dataset to train the different standard CNN architectures. To create computer vision algorithms, researchers created the ImageNet dataset, a massive collection of different types of images. It has about 1.4 million labeled images from 1,000 different classes, making it the most comprehensive and extensive image dataset. There is a significant difference between ImageNet data and specific medical imaging data (mammograms). In contrast to the typical ImageNet classification system of 1,000 classes, medical image classification tasks often include fewer classes (three classes for diagnosing breast cancer, *i.e.,* normal, benign, and malignant).

The standard CNN models accomplish transfer learning tasks. The models, like ResNet50 Inception V3 MobileNet, are multi-layer architectures pre-trained on ImageNet. The earlier layers of the pre-trained model extract general features like edges and texture. Higher layers extract features specific to the data on which the model is trained. So, the design of these pre-trained models is probably not ideal for medical imaging tasks because traditional ImageNet architectures have a significant number of parameters concentrated at the higher layers (*Raghu et al., 2019*).

## Analysis of existing techniques for breast cancer detection and classification

Despite exceptional advancements in examinations and patient administration, breast cancer remains the second most recognized disease and the chief reason for cancer casualties among women in Pakistan. As per the report of the International Agency of Research, there were 34,066 women recently diagnosed with breast cancer in 2018

(*Khan et al., 2021*). Pakistan has the highest ratio of breast cancer cases in all of Asia, one in nine. It is an alarming situation for our beloved country, Pakistan.

Machine learning (ML) frameworks have played a significant part in detecting numerous diseases. Researchers have used ML algorithms to get accurate results in minimum time. Later, deep learning overcame ML. DL is an ML subclass based on ANN in which multiple layers extract high-level features from data. However, the efficient DL models are data-hungry and complex and require training on thousands or even millions of images for valuable results (*Houssein, Emam & Ali, 2022*). Such a considerable amount of data is unavailable in most medical imaging modalities.

The transfer learning technique got the extensive focus of research communities to overcome the problem. Transfer learning is a deep learning strategy that involves moving knowledge from one model to another. In this technique, the previous knowledge of the pre-trained model can be utilized to solve a new but relevant problem (*Khamparia et al., 2021*). The primary idea behind transfer learning is to use what has been learned from one domain to enhance generalization for another but similar domain without retaining the model. With the help of TL, valuable results can be obtained on a small dataset. Moreover, computational costs and training time can also be reduced with the help of transfer learning (*Hosna et al., 2022*).

Table 1 elaborates on the existing literature on breast cancer detection and analyzes the deep learning and transfer learning studies. It also gives information about the convolution base of pre-trained CNN in different transfer learning strategies.

The researchers used the complete convolutional base to extract features from different datasets in all the studies. The higher layers of the pre-trained model produce deep-level features specific to the training dataset. It has been observed that the performance of transfer learning schemes may not be optimal with the dataset-specific features. Some studies adopt the fine-tuning strategy to train some layers of the CNN model on the selected dataset. The fine-tuning strategy can improve the results because the model is trained on the desired dataset. However, fine-tuning is complex, time-consuming, and requires more computational power and resources. Table 1 presents an analysis of the existing approaches for BCDC using DL and transfer learning models. It compares the existing studies in terms of accuracy, use of GFs, DSFs, convolution base (CB), and fine-tuning (FT) of the models, as well as the redesigned new classifier (NC). The above literature analysis concludes that transfer learning gives more accurate results with smaller datasets. The problem of overfitting also occurs in DL because it requires massive data for better results. Additionally, the DL requires more computational powers and training time. The complexity of the hyperparameters makes it difficult to achieve better outcomes in DL (*Houssein, Emam & Ali, 2022*).

A DL model's performance on a limited dataset, such as medical images, can be enhanced with the help of transfer learning. It can reduce training time and computational costs (*Hosna et al., 2022*). So, this study has designed an improved transfer learning strategy that has detected and classified breast cancer more efficiently. The convolution base of the pre-trained CNN model has been studied. It has also examined the effect of dataset-specific and generic features on classification results.

Safdar Ali Khan et al. (2024), *PeerJ Comput. Sci.*, DOI 10.7717/peerj-cs.1938

**Table 1** Analysis of existing studies.

| Ref | Dataset | Image size | Model | Accuracy | GF | DSF | Full CB | FT | NC | MU | CPU | RT |
|---|---|---|---|---|---|---|---|---|---|---|---|---|
| Aljuaid et al. (2022) | BreakHis | 7,909 | Inception V3 | 99.70 | ✓ | ✓ | ✓ | × | × | – | – | – |
| | BreakHis | 1,153 | ResNet 18 | 99.70 | ✓ | ✓ | ✓ | × | × | – | – | – |
| | BreakHis | 1,153 | ShuffleNet | 99.70 | ✓ | ✓ | ✓ | × | × | – | – | – |
| Jabeen et al. (2022) | BUSI | 1,153 | DarkNet-53 | 99.10 | ✓ | ✓ | ✓ | ✓ | ✓ | – | – | ✓ |
| | Mendeley | 1,153 | EfficientNetB2 | 99.00 | ✓ | ✓ | ✓ | × | ✓ | – | – | – |
| | Mendeley | 410 | InceptionV3 | 99.00 | ✓ | ✓ | ✓ | × | ✓ | – | – | – |
| Ayana et al. (2022) | Mendeley | 7,909 | ResNet50 | 99.00 | ✓ | ✓ | ✓ | × | ✓ | – | – | – |
| | MT-U | – | EfficientNetB2 | 99.00 | ✓ | ✓ | ✓ | × | ✓ | – | – | – |
| | MT-U | 9,927 | InceptionV3 | 99.00 | ✓ | ✓ | ✓ | × | ✓ | – | – | – |
| | MT-U | 7,909 | ResNet50 | 99.00 | ✓ | ✓ | ✓ | × | ✓ | – | – | – |
| | BreakHis | 7,909 | DenseNet | 98.90 | ✓ | ✓ | ✓ | ✓ | ✓ | – | – | – |
| Zheng et al. (2023) | BreakHis | 780 | ResNet | 98.90 | ✓ | ✓ | ✓ | ✓ | ✓ | – | – | – |
| | BreakHis | 7,909 | VGG16 &19 | 98.90 | ✓ | ✓ | ✓ | ✓ | ✓ | – | – | – |
| | BreakHis | 7,909 | Xception | 98.90 | ✓ | ✓ | ✓ | ✓ | ✓ | – | – | – |
| Houssein, Emam & Ali (2022) | CBIS-DDSM | 7,909 | IMPA-ResNet50 | 98.00 | ✓ | ✓ | ✓ | × | ✓ | – | – | – |
| | MIAS | 7,909 | IMPA-ResNet50 | 98.00 | ✓ | ✓ | ✓ | × | ✓ | – | – | – |
| | MIAS | 322 | InceptionV3 | 98.00 | ✓ | ✓ | ✓ | × | ✓ | – | – | – |
| Saber et al. (2021) | MIAS | 322 | ResNet50 | 98.00 | ✓ | ✓ | ✓ | × | ✓ | – | – | – |
| | MIAS | 322 | VGG16/19 | 98.00 | ✓ | ✓ | ✓ | × | ✓ | – | – | – |
| | DDSM | – | MobileNet | 97.91 | ✓ | ✓ | ✓ | × | ✓ | – | – | – |
| | CBIS-DDSM | – | AlexNet | 97.90 | ✓ | ✓ | ✓ | × | ✓ | – | – | – |
| Tan et al. (2023) | CBIS-DDSM | – | Inception V3 | 97.90 | ✓ | ✓ | ✓ | × | ✓ | – | – | – |
| | CBIS-DDSM | – | ResNet50 | 97.90 | ✓ | ✓ | ✓ | × | ✓ | – | – | – |
| | CBIS-DDSM | – | VGG16/19 | 97.90 | ✓ | ✓ | ✓ | × | ✓ | – | – | – |
| Khan et al. (2019) | BreakHis | – | GoogLeNet VGGNet | 97.50 | ✓ | ✓ | ✓ | × | ✓ | – | – | – |
| | BreakHis | – | ResNet50 | 97.50 | ✓ | ✓ | ✓ | × | ✓ | – | – | – |
| Ayana et al. (2023) | DDSM-M | – | PVT | 95.00 | ✓ | ✓ | ✓ | ✓ | ✓ | – | – | – |
| | DDSM-M | – | Swin | 95.00 | ✓ | ✓ | ✓ | ✓ | ✓ | – | – | – |
| | DDSM-M | – | ViT-base | 95.00 | ✓ | ✓ | ✓ | ✓ | ✓ | – | – | – |
| Sharma & Mehra (2020) | BreakHis | — | ResNet50 | 93.97 | ✓ | ✓ | ✓ | × | ✓ | – | – | – |
| | BreakHis | – | SVM | 93.97 | ✓ | ✓ | ✓ | × | ✓ | – | – | – |
| | BreakHis | 2,500 | VGG16/19 | 93.97 | ✓ | ✓ | ✓ | × | ✓ | – | – | – |

Safdar Ali Khan et al. (2024), *PeerJ Comput. Sci.*, DOI 10.7717/peerj-cs.1938

**Table 1** (*continued*)

| Ref | Dataset | Image size | Model | Accuracy | GF | DSF | Full CB | FT | NC | MU | CPU | RT |
|---|---|---|---|---|---|---|---|---|---|---|---|---|
| | CBIS-DDSM | – | MobileNetV2 | 93.50 | ✓ | ✓ | ✓ | ✓ | ✓ | – | – | ✓ |
| | CBIS-DDSM | – | NasNet | 93.50 | ✓ | ✓ | ✓ | ✓ | ✓ | – | – | ✓ |
| *Zahoor, Shoaib &* | Inbreast | | MobileNetV2 | 93.50 | ✓ | ✓ | ✓ | ✓ | ✓ | – | – | ✓ |
| *Lali (2022)* | Inbreast | 7,909 | NasNet | 93.50 | ✓ | ✓ | ✓ | ✓ | ✓ | – | – | ✓ |
| | MIAS | 7,909 | MobileNetV2 | 93.50 | ✓ | ✓ | ✓ | ✓ | ✓ | – | – | ✓ |
| | MIAS | 7,909 | NasNet | 93.50 | ✓ | ✓ | ✓ | ✓ | ✓ | – | – | ✓ |
| | Kaggle-H | —— | DenseNet | 93.00 | ✓ | ✓ | ✓ | ✓ | ✓ | – | – | – |
| | Databiox | – | DenseNet | 92.50 | ✓ | ✓ | ✓ | × | ✓ | – | – | – |
| *Sujatha et al.* | Databiox | – | InceptionReNet | 92.50 | ✓ | ✓ | ✓ | × | ✓ | – | – | – |
| *(2023)* | Databiox | – | VGG16 | 92.50 | ✓ | ✓ | ✓ | × | ✓ | – | – | – |
| | Databiox | – | VGG19 | 92.50 | ✓ | ✓ | ✓ | × | ✓ | – | – | – |
| *Yadavendra &* | BreakHis | 922 | Xception | 90.00 | – | – | – | × | – | – | – | – |
| *Chand (2020)* | | | | | | | | | | | | |
| *Kumar & Rao* | Histp. Images | 7,909 | CNN | 90.00 | – | – | – | – | – | – | – | – |
| *(2018)* | | | | | | | | | | | | |
| *Alruwaili &* | MIAS | 7,909 | MOD-ResNet50 | 89.50 | ✓ | ✓ | ✓ | × | ✓ | – | – | – |
| *Gouda (2022)* | MIAS | 7,909 | Nasnet-Mobile | 89.50 | ✓ | ✓ | ✓ | × | ✓ | – | – | – |
| | DDSM | 7,909 | AlexNet | 88.30 | ✓ | ✓ | ✓ | ✓ | ✓ | – | – | – |
| *Khamparia et al.* | DDSM | – | MobileNet | 88.30 | ✓ | ✓ | ✓ | ✓ | ✓ | – | – | – |
| *(2021)* | DDSM | – | Modified VGG | 88.30 | ✓ | ✓ | ✓ | ✓ | ✓ | – | – | – |
| | DDSM | – | ResNet 50 | 88.30 | ✓ | ✓ | ✓ | ✓ | ✓ | – | – | – |
| | DDSM | – | VGG16/19 | 88.30 | ✓ | ✓ | ✓ | ✓ | ✓ | – | – | – |
| *Khalil et al.* | H-IDC | – | 3D U-Net | 87.00 | ✓ | ✓ | ✓ | ✓ | × | – | – | – |
| *(2023a)* and | | | | | | | | | | | | |
| *Khalil et al.* | | | | | | | | | | | | |
| *(2023b)* | | | | | | | | | | | | |
| *Sreeraj & Joy* | MITOS-14 | – | YOLO-V3 | 87.00 | ✓ | ✓ | ✓ | × | ✓ | – | – | – |
| *(2021)* | MITOS-14 | – | Tiny-YOLO | 85.00 | ✓ | ✓ | ✓ | × | ✓ | – | – | – |
| | BCDR | – | DenseNet | 84.00 | ✓ | ✓ | ✓ | × | ✓ | – | – | ✓ |
| *Azevedo, Silva &* | BCDR | – | Quantum TL | 84.00 | ✓ | ✓ | ✓ | × | × | – | – | ✓ |
| *Dutra (2022)* | BCDR | – | ResNet | 84.00 | ✓ | ✓ | ✓ | × | ✓ | – | – | ✓ |
| | BCDR | – | RexNetXt | 84.00 | ✓ | ✓ | ✓ | × | ✓ | – | – | ✓ |
| *Lu, Loh & Huang* | TaiwanHospital | – | CNN BI-RADS | 82.00 | – | – | – | – | – | – | – | – |
| *(2019)* | | | | | | | | | | | | |
| *Sreeraj & Joy* | MITOS-14 | – | YOLO-V1 | 80.00 | ✓ | ✓ | ✓ | × | ✓ | – | – | – |
| *(2021)* | MITOS-14 | – | YOLO-V2 | 77.00 | ✓ | ✓ | ✓ | × | ✓ | – | – | – |

**Notes.**
GF, Generic Features; DSF, Domain Specific Features; CB, Convolution Base; FT, Fine Tuning; NC, New Classifier; MU, Memory Utilization; CPU, Central Processing Unit; RT, Response Time; ✓, Evaluate in study; ×, Not Evaluated in Study; –, Not Considered.

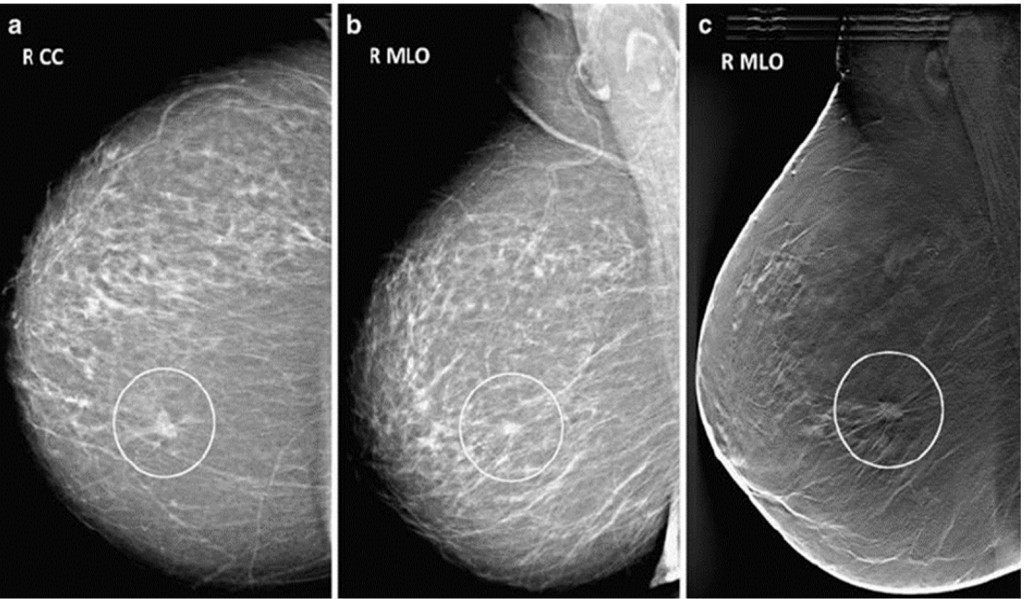

**Figure 3   Mammogram CC and MLO views of stage 1 breast cancer.**

## MATERIALS & METHODS

The proposed methodology in this article falls into the feature-based transfer learning category as this study has extracted features from the initial layers of the source model and utilized its feature representation in our target domain. Generic features from the earlier layers of the source model that contain more basic information about the different objects that can be transferred to a new target task have been extracted. This approach can improve performance by reusing the different generic feature representations learned from the source dataset of the pre-trained model.

Usually, in most of the research, the researchers used the entire convolution base of the CNN models for feature extraction in transfer learning (*Houssein, Emam & Ali, 2022*). They were used to remove the classifier part of the selected CNN model and extract features with all layers of the convolution base of the model, as shown in Fig. 3. Fine-tuning has also been performed on a few layers to make the CNN model more appropriate for the specific task. However, fine-tuning is a time-consuming and costly process in transfer learning.

### Generic feature-based transfer learning strategy

Traditional CNN models have many parameters concentrated at the higher layers. As the CNN models are pre-trained on ImageNet, the higher layers of the convolution base are concentrated on dataset-specific features. In comparison, the lower layers extract general features like edges and texture. However, according to *Raghu et al. (2019)*, earlier layers are more beneficial for downstream tasks than the complete architecture of the pre-trained model. This article proposes an improved transfer learning strategy named generic features-based transfer learning, which utilizes the generic features extracted from

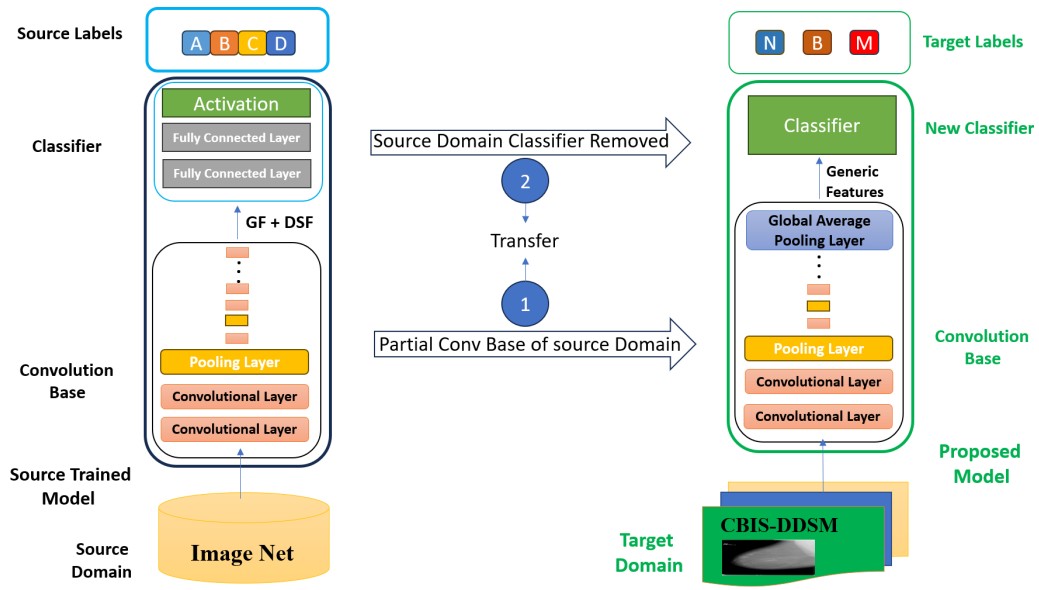

**Figure 4** Generic features-based transfer learning (GFTL) strategy.

earlier layers of the model to accomplish transfer learning tasks with better accuracy and reduced complexity.

In GFTL, the higher layers of the pre-trained model are discarded (as shown in Fig. 4) because these are not beneficial in new transfer learning tasks. Different earlier model layers have been used to extract generic features like edges, basic textures, and simple shapes. The classifier part of the model has also been replaced with a new classifier and trained on the new dataset to classify that data. Generic features (GF) refer to fundamental visual patterns unrelated to a specific object. The network learns generic features in the first few layers of a pre-trained CNN. These features capture general visual patterns such as edges, basic textures, and simple shapes. These generic features are fundamental and not specific to a specific shape of an object. These earlier layers can see simple generic features irrespective of their spatial location within the image. A first-layer edge detector can detect edges regardless of their positioning within the image, whether in the center or the corners.

Domain-specific features (DSF) refers to complex visual patterns related to an object's shape. The higher layers of a CNN are responsible for extracting high-level domain-specific features. Recognizing individual objects or parts of an object within an image is often the job of the higher layers of a pre-trained CNN. The higher layers capture domain-specific features like discrete patterns, item details, and contextual relationships in the input image. These domain-specific features can be trained to identify objects such as followers, dogs, cats, automobiles, and other similar entities. The high-level features are substantially more abstract, semantically meaningful, and task-specific than the generic low-level features produced by the pre-trained CNN's early layers.

The GFTL scheme consists of five major phases: image acquisition, image preprocessing, selection of the pre-trained CNN model, feature extraction from different layers of the

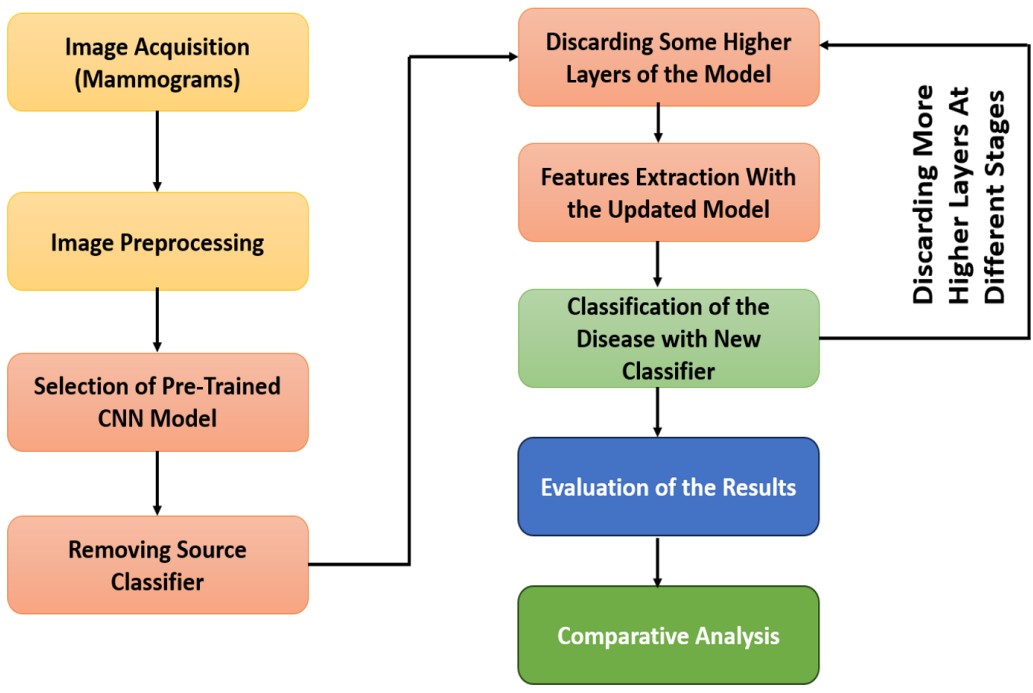

**Figure 5** **Workflow diagram of GFTL strategy.**

convolution base of the pre-trained CNN model, and classification of breast cancer. The flow diagram of our proposed methodology can be shown in Fig. 5.

## Dataset and image acquisition

As discussed earlier, mammography has widely been considered a gold standard for breast cancer detection; this article has also focused on mammogram images obtained from the reputable dataset, *i.e.,* the curated breast imaging subset of the Digital Database for Screening Mammography (CBIS-DDSM) created by National Cancer Institute (NCI). This dataset is the subset and updated form of the Digital Database for Screening Mammography (DDSM). The dataset was curated by gathering and merging digital mammography images from several sources, including multiple hospitals and healthcare institutions. The medical experts conducted a comprehensive analysis of the cases to label and classify the mammograms accurately, detecting the presence of masses, calcifications, or any other anomalies. It is a well-managed, latest, authenticated, and publicly available dataset of mammograms used in several of the latest studies (*Zahoor, Shoaib & Lali, 2022*). It contains 1,589 mammogram images of three classes, *i.e.,* normal, benign, and malignant. There are two standard mammogram views: the craniocaudal (CC) view and the medio lateral oblique (MLO). All the images are selected and verified by a skilled mammographer. The dataset has been divided into two folders. The first folder contains 1,229 images that are used in this study. The selected dataset has been preprocessed and then automatically divided into testing and training parts through the sklearn train-test-split function.

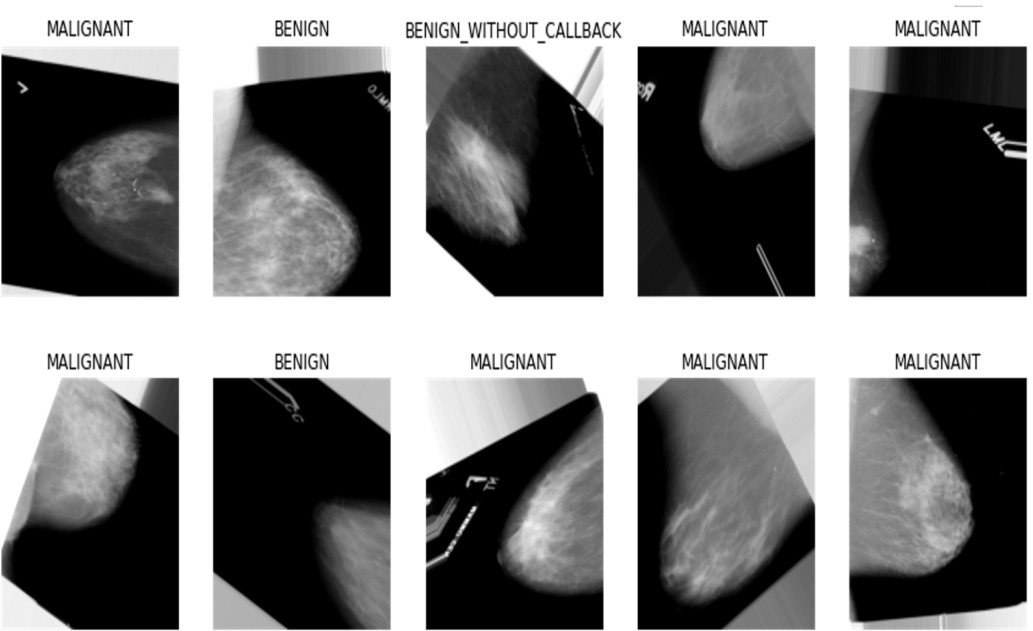

Figure 6   Output of data augmentation process.

## Image preprocessing

The visibility of images is an essential aspect of getting valuable accuracy. In the dataset, few images have noise like salt and pepper noise that have been removed through median filter. Sometimes, the cancerous regions in images may be blurred. The blurriness has also been removed from the images using the Gaussian filter. Resizing has been performed to make all the images of a consistent size and to save computational resources. All the images have been resized to a constant size of $224 \times 224$.

## Data augmentation

As discussed in the earlier chapter, the problem of scarce data is common in medical image datasets. So, data augmentation is an outstanding technique to overcome this issue, as it is used to increase the number of images by generating modified copies of the original images (*Ayana et al., 2023*). This study has adopted the data augmentation technique of TensorFlow library to increase the dataset mentioned, as shown in Table 1. Applied augmentation techniques are described as follows, and the results are shown in Fig. 6.

- Flip-up-down This technique flips the original image up and down to create a new copy of the image. Since every image can be represented as a matrix of pixels, each pixel stores its distinct information. So, the "up-down flipping" process involves the vertical reversal of rows and columns inside a matrix, creating new rows and columns to obtain a new image.
- Flip-left–right This technique flips the original image from left to right to create a new copy of the image. The "left–right flipping" process involves horizontally reversing rows

**Table 2   Details of the dataset used in this article.**

| Type of Mammogram | No. of original images | No. of images after data augmentation process |
|---|---|---|
| Normal | 83 | 664 |
| Benign | 549 | 2,196 |
| Malignant | 597 | 2,388 |
| Total: | 1,229 | 5,248 |

and columns inside the pixel's matrix, creating new rows and columns to obtain a new image.

- Adjust-brightness. This technique adjusts the brightness of the original image to enhance the spatial information of the image and creates a new copy of the image. It is a color augmentation technique in which a random value is "added" to each pixel of the image matrix, creating a new image with a different brightness.
- Adjust-contrast. This technique adjusts the contrast of the original image to create variations in the grayscale image and make a new copy of the image. It is also a color augmentation technique in which a random value is multiplied by each pixel of the image matrix, creating a new image with a different contrast.
- Rotate90. This technique is used to rotate the original image by 90 degrees to create a new copy of the image. The geometric augmentation technique rotates the pixels of the image matrix by different angles to create a new image. This study has rotated the images by 90 degrees.

## Train-test split

The train-test split technique of the sklearn library is also a good technique used to divide data into training and testing parts automatically. The details of the data set are shown in Table 2. After performing the data augmentation technique, this article used and divided the data at an 80–20 ratio of training and testing parts. Image shuffling is also performed, which involves randomly dividing images into training and testing parts.

## ImageNet pre-trained model selection

The CNNs significantly advance computer vision, enabling machines to identify objects and images with new accuracy levels. Many CNN models are available as part of the Keras library in Python. Some popular and recent CNN models are ResNet50, MobileNet, AlexNet, Inception V3, and LeNet. Every model has strengths and a brilliant record in image classification. These standard CNN architectures pre-trained on ImageNet are commonly used to accomplish transfer learning tasks (*Khalil et al., 2023b*). Every CNN model has millions of trainable parameters trained on the ImageNet dataset of 1,000 different classes. The experiments have been conducted on the popular pre-trained CNN model ResNet50.

The ResNet50 model shows an outstanding improvement among many CNN models and has emerged as a significant deep-learning breakthrough. It is well-known for its fifty-layer design, surpassing the number of layers in its ancestors and hence achieving

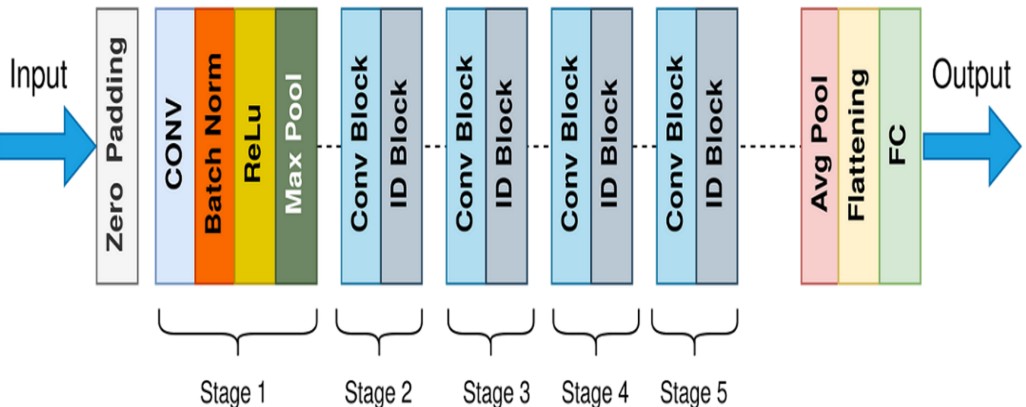

**Figure 7 General architecture of ResNet50.**

performance in models with a shorter depth. The ability of the model above design to handle the "vanishing gradient problem effectively" constitutes a significant advancement in artificial intelligence (*Roodschild, Gotay & Will, 2020*; *Thapa et al., 2022*). The depth to which typical CNN can be successfully trained is constrained by the above-stated issue.

ResNet is the short form of the residual network first introduced in 2015 by *He et al. (2016)*. The architecture of ResNet50 contains a few stages: the initial convolutional layer (ICL), the residual blocks (RB), the global average pooling layer (GAPL), and the fully connected layer (FCL), respectively. The utilization of residual blocks is the fundamental concept behind this model. Each block of residual contains two convolutional layers. That block also contains a Rectified Linear Unit (ReLU) activation function and batch normalization layer. The general architecture of ResNet50 is shown in Fig. 7. These blocks enable the network to bypass specific layers during the training process. Because of this methodology, it is now possible to train significantly deeper networks than before without losing accuracy. An incredibly deeper 152-layer model was trained as an outcome of the residual units and won the LSVRC competition in 2015.

The ResNet50 model is frequently used in transfer learning due to its exceptional performance and adaptability (*Hossain et al., 2022*; *Reddy & Juliet, 2019*; *Saber et al., 2021*). It has demonstrated exceptional performance in computer vision tasks, such as picture segmentation, object recognition, and image classification (*Kasaei, 2021*). The model's deep architecture and use of residual blocks enable it to accurately perceive tiny details in images and demonstrate better generalization abilities when faced with novel data (*Krishna & Kalluri, 2019*). Skip connections help the gradients flow smoothly throughout the training phase, enabling deep network progress.

The vital benefit of this neural network is its improved performance in consistently classifying big data sets of images like ImageNet. Performance-wise, ResNet50 outperformed earlier state-of-the-art models. Only 6.71 percent of the top 5 errors were made in the 2015 ImageNet large-scale visual recognition challenge (*He et al., 2016*).

This architecture demonstrated the effectiveness of residual architecture, enabling the construction of neural networks with more efficacy and accuracy.

## Convolution base of CNN models

The standard pre-trained CNN model has two main parts, *i.e.,* the convolutional and classifier parts. The convolution part contains convolutional layers that extract features from the input images. The classifier part is used to classify the input images into different classes according to learned information from the convolutional part. The original classifier of the model is trained to classify 1,000 different classes of the ImageNet dataset. However, it is necessary to classify images into three classes, *i.e.,* normal, benign, and malignant. So, the classifier part is removed, and only the convolution base has been used for feature extraction.

## Features extraction from different layers of convolution base

The ability of CNN models to extract distinctive features holds significant importance in computer vision applications, enabling computers to interpret, analyze, and respond to visual data. Specific ML techniques, such as transfer learning, are used to leverage data from one task or domain to enhance performance in other correlated tasks or domains. Feature extraction plays a significant role in the domain of transfer learning. In transfer learning, feature extraction refers to utilizing the learned features from a pre-trained model as the foundation for a subsequent task.

A relatively small dataset with fewer parameters is selected compared to ImageNet; the selected pre-trained model design is probably not ideal for our task because traditional CNN models have a significant number of parameters concentrated at the higher layers. As the ResNet50 model had been pre-trained on ImageNet, the higher layers are concentrated on dataset-specific features in the convolution base. In comparison, the lower layers extract general features like edges and texture.

Therefore, some higher layers of the convolution base of the pre-trained model ResNet50 are discarded by using the layer method of the TensorFlow library. Instead, different earlier layers have been used to extract features. The main idea of this study is to determine the effects of feature extraction at different levels of the convolution based on the accuracy, as shown in Fig. 4. To extract more generic features, some higher layers are discarded at different iterations and compared to the results of the entire and modified convolution bases of the model ResNet50. The GAPL is used at the top of the convolution base to get a fixed-size feature map for each feature channel.

## Breast cancer classification

Classification is an essential step in the process of transfer learning, which is a subfield of machine learning and deep learning. It is also an essential component in many different applications of disease diagnosis. During the training part of the classification, a set of patterns and features are learned from a set of data that has already been labeled. These patterns and features are then used to label new data. The first step is to get labeled data, where every image is put into a particular group or set of classes. To make an effective and

better model, you need a large set of data that is both complete and representative. Results from a different validation dataset evaluate the model's efficacy.

Breast cancer classification is the last step of our proposed methodology. The proposed methodology has adopted the multiclass classification method to classify the input images according to the selected features. This step classified images/frames into normal, benign, and malignant portions. Final experimental results were obtained using the extracted features of images we collected during the abovementioned step. Features of the images have been compared with the previously calculated values. The extracted features by the transferred model have been input to the classifier's input layer.

A new deep neural network classifier is designed for BCDC. The classifier has three layers, *i.e.,* the input layer with 1,024 neurons with the ReLu activation function, hidden layers with 512 and 256 neurons with the ReLu activation function, and the output layer with a SoftMax activation function. The SoftMax activation function of the output layer has classified the mammogram images into three categories, *i.e.,* Normal, Benign, and Malignant. The classifier has been compiled with the RMSprop optimization function with a learning rate of 0.0001. It has used the category-cross-entropy loss function to judge the model's performance. The model has been trained for 400 epochs with a batch size of 32.

## Evaluation of the results

The evaluation of a model is the crucial component of creating an efficient model. Regular evaluations must be conducted during training to ensure the CNN model is learning and improving. Evaluating the CNN model as it is being trained provides insight into its progress and helps pinpoint any issues that may need to be addressed. If the model is analyzed while trained, fixing problems like over-fitting and under-fitting early on can boost the model's accuracy.

The performance of CNN models is evaluated using a set of functions called metrics. The most common classification evaluation measures, such as accuracy, precision, recall, F1-score, and resource efficiency, are used to highlight the performance of the proposed research work. These evaluation measures are the built-in metrics that the TensorFlow library provides. The transferred model's predictions (correct/wrong) are analyzed and compared with actual values in the breast cancer dataset through the measures. Furthermore, the model's efficiency is measured in terms of training cost and time through resource efficiency. The model accuracy ($A_m$) is the ratio between the true outcomes to the total measurements as shown in Eq. (1). The true classifications (TC) consist of true positive ($T_P$) and true negative ($T_N$) outcomes, and the total measurements (TM) are the sum of both the true and false outcomes.

$$A_m = \frac{TC}{TM} = \frac{T_p + T_n}{(T_p + T_n) + (F_p + F_n)}. \tag{1}$$

The model precision is the ratio between the true positive cases ($T_p$) and total positive (P) outcomes, as shown in Eq. (2).

$$P_m = \frac{T_p}{P} = \frac{T_p}{(T_p + F_p)}. \tag{2}$$

The model recall is a measure that represents the true positive rate of the model, which is the ratio between true positive outcomes and the sum of true positive and false negative outcomes, as shown in Eq. (3).

$$R_m = \frac{T_p}{(T_p + F_n)}.$$  (3)

The F1 score of the model ($F1_m$) is another metric that combines precision and recall using their harmonic mean and maximizing the $F1_m$ score indicates simultaneous maximization of precision and recall, as shown in Eq. (4).

$$F1_m = \frac{P_m * R_m}{(P_m + R_m)}.$$  (4)

When solving a classification problem, the confusion matrix is a standard method to analyze the model's performance. This strategy can be used for both binary and multiclass classification problems. The confusion matrix, also known as the error matrix, is a numerical matrix that provides insight into the instances where a model shows confusion. The confusion matrix is a logical representation of the predictive accuracy of a classification model. It illustrates the distribution of predictions across different classes, allowing for a precise mapping between the model's predictions and the original class labels of the data. Several evaluation metrics used to evaluate the CNN model's performance can be computed with the help of a confusion matrix.

The number of classes determines the size of the confusion matrix, which is displayed as a square matrix. The columns provide the predicted classes of the model, whereas the rows indicate the actual observed classes in the dataset. The matrix's diagonal elements represent the places where the anticipated label and the actual label coincide. On the other hand, the off-diagonal elements display the number of instances when the classifier incorrectly predicted the label. The confusion matrix is used to demonstrate the classifier's performance based on the values $T_P$, $F_P$, $T_N$, and $F_N$. The above values are plotted against each other to show a confusion matrix.

## RESULTS AND DISCUSSION

As discussed in earlier sections, the convolution base of the CNN model has been evaluated, and the impact of different higher layers on classification results has been noticed. The impact of DSFs extracted from convolution base higher layers (CBHL) and GFs extracted from initial convolution base (ICB) on classification results has been analyzed in the following discussion. The evaluation of the GFTL is carried out in four stages, each of which employs a specific discard level of the convolution base. The classifier is trained for 400 epochs with a batch size of 32 with root mean squared propagation optimization function with a 0.0001 learning rate. The performance of the GFTL is evaluated on measures like accuracy, precision, recall, F1-score, and time complexity.

### Stage 1—using the full convolution base of the CNN model

In this first implementation stage, the features are extracted using the full convolution base (FCB) of the model ResNet50. The FCB of the model has 175 layers with a GAPL at the

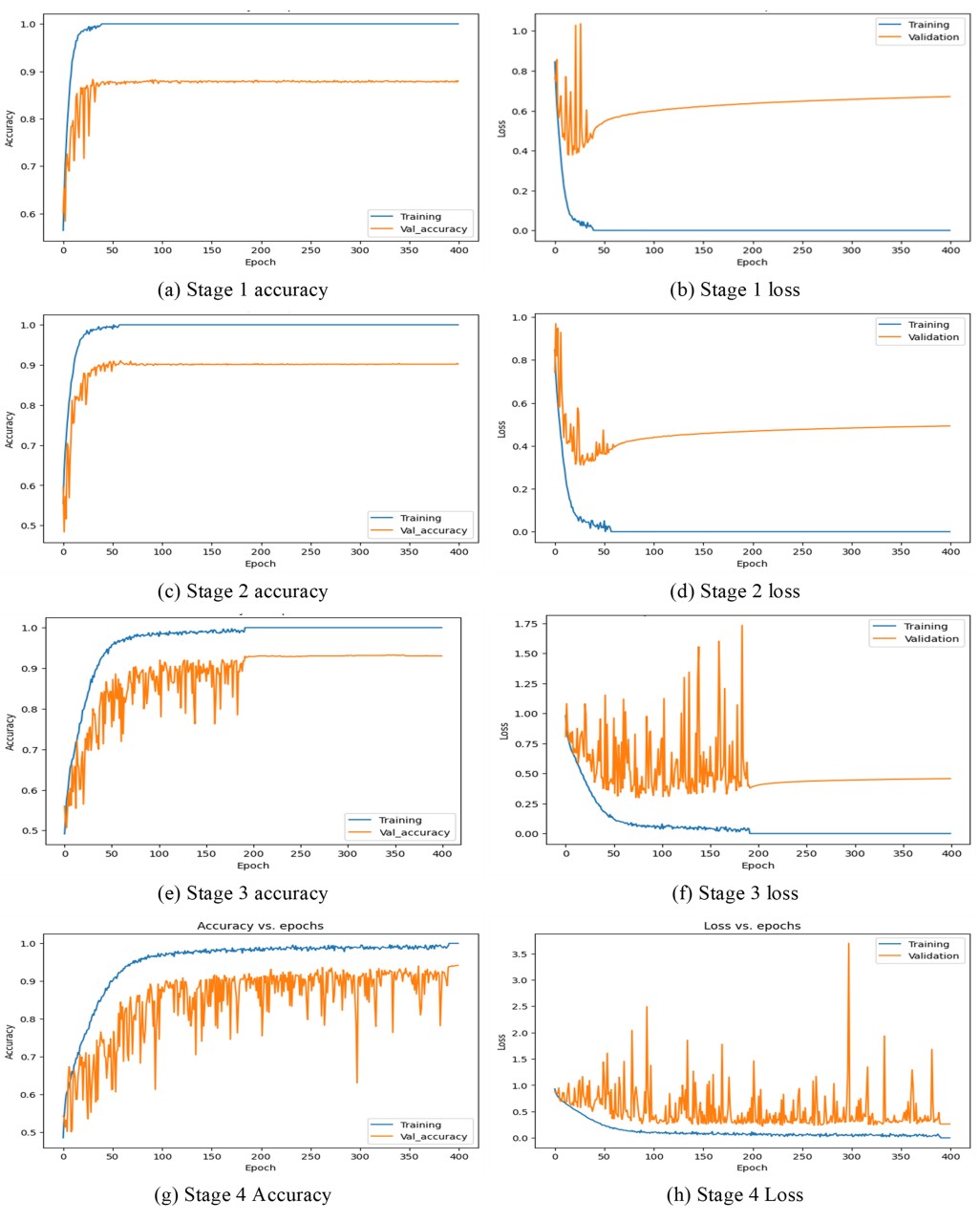

**Figure 8   Accuracy and loss at different stages.**

top, and the model has a total of 2,35,87,712 parameters. Using features extracted from the FCB of the ResNet50 model, the classifier has achieved 100% training accuracy and 88% validation accuracy. The accuracy and loss of the model are shown in Figs. 8A and 8B, respectively. Accordingly, in stage 1, the precision of the model is observed as 88.80%, with recall at 87.98% and an F1 score of 88.38%. The corresponding confusion matrix is shown in Fig. 9A)

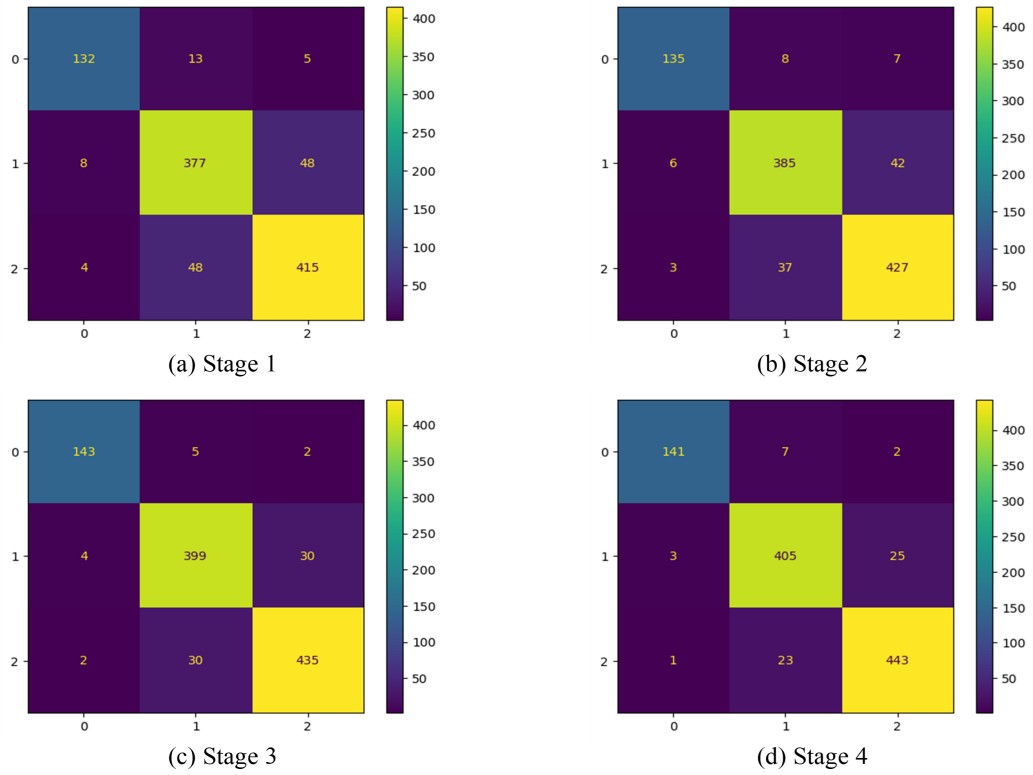

**Figure 9** Confusion matrix at different stages.

*Stage 2—discarding 10 CBHL*

In the 2nd stage of the implementation stage, the features are extracted from the modified model after discarding the ten CBHLs. After modification of the transferred model, it has 165 layers, a GAPL at the top, and a total of 1,91,15,904 parameters. Using features extracted by the modified convolution base of the ResNet50 model at 2nd stage, the classifier achieved 100% training accuracy and 90.19% validation accuracy. The accuracy and loss of the model are shown in Figs. 8C and 8D, respectively. Accordingly, the precision of the model is 91%, the recall is 90.12%, and the f1-score of our model is 90.54%. The respective confusion matrix also demonstrates the model's performance, as shown in Fig. 9B.

*Stage 3: discarding 21 CBHL*

In the 3rd stage of the implementation, the 21 CBHLs are discarded, and features are extracted from the modified model. After modification of the transferred model, the model has 154 layers and a GAPL at the top. With 155 layers, the model has a total of 14,6,44,096 parameters. Using features extracted by the modified convolution base of the ResNet50 model, the classifier achieved 100% training accuracy and 93.04% validation accuracy. The accuracy and loss of the model in stage 3 are shown in Figs. 8E and 8F, respectively. The precision of the model is observed as 93.69%, with a recall of 93.55% and an F1 score of 93.61% at stage 3. The corresponding confusion matrix is shown in Fig. 9C.

**Table 3  Performance metrics at different discard levels.**

| DSSF Level | Measured values (%) | | | |
|---|---|---|---|---|
| | Accuracy | Precision | Recall | F1-Score |
| Full | 88% | 88.80% | 87.98% | 88.38% |
| 10 | 90.19% | 91% | 90.12% | 90.54% |
| 21 | 93.04% | 93.69% | 93.54% | 93.61% |
| 32 | 94.19% | 94.87% | 94.13% | 94.49% |

***Stage 4: discarding 32 CBHL***

In the 4th stage, the 32 CBHLs are discarded, and then features from the modified model are extracted. With the above discard level of CBHL, the model has 143 layers and a GAPL at the top. With 144 layers, the model has a total of 8,5,89,184 parameters. Using features extracted by the modified convolution base of the ResNet50 model at stage 4, the classifier achieved 100% training accuracy and 94.19% validation accuracy. The accuracy and loss of the model are shown in Figs. 8G and 8H, respectively. Accordingly, the model's precision is 94.87%, with a recall of 94.13% and an F1 score of 94.49%. The corresponding confusion is shown in Fig. 9D.

## Comparison and analysis of results

In this section, the comparative analysis of all the stages is presented. The model's performance with respect to accuracy, recall, precision, and F1-score has been demonstrated in Table 3. By discarding 10, 21, and 32 higher layers of the convolution base, the accuracy of the CNN model ResNet50 is increased to 90.19%, 93.04%, and 94.19%, respectively, and features extraction time also decreased by the proposed methodology as the significant number of parameters decreased by discarding higher layers.

It can be seen in Fig. 10 that with different levels of CBHL discard, the model's performance has gradually increased. After discarding 32 layers, it has achieved the most optimal results.

## K-fold cross-validation

The stratified K-fold cross-validation is also performed on ten different subsets of data to validate the model's performance. Cross-validation is performed on the original model (with full convolution base) and modified model (with 32 layers discarded) to compare their performance. The results show that the modified model has better accuracy than the original model, as shown in Table 4.

## Analysis of computational requirements offloading

Discarding CB at different levels affects the performance metrics and computational requirements. The previous subsection analysed the model performance on four different levels, and it was observed that DSF discarding level 32 improves the model accuracy. The analysis of computational requirements for different phases of the model flow is shown

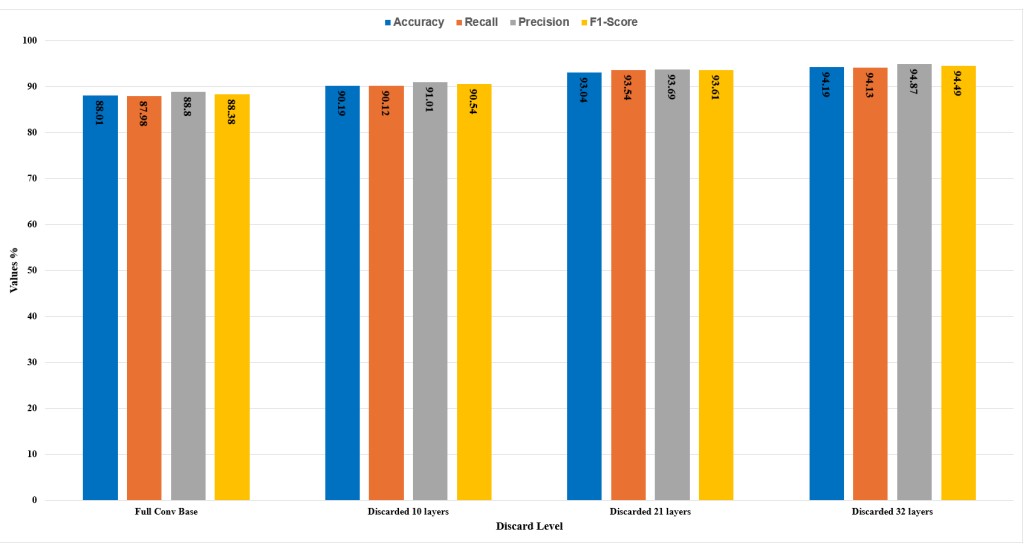

**Figure 10** Comparative analysis of the results.

**Table 4** Stratified K-fold cross-validation results.

| Validation split | Validation accuracy of original model (with full conv base) | Validation accuracy of modified model (with 32 layers discarded) |
|---|---|---|
| 1 | 89.52 | 95.23 |
| 2 | 84.76 | 92.38 |
| 3 | 91.42 | 94.28 |
| 4 | 90.47 | 91.42 |
| 5 | 86.66 | 89.52 |
| 6 | 84.76 | 92.38 |
| 7 | 87.61 | 94.28 |
| 8 | 83.8 | 95.23 |
| 9 | 91.42 | 97.14 |
| 10 | 85.71 | 95.24 |
| Average validation results | 87.62 | 93.71 |

in Fig. 11 and Table 5. Experiments were conducted with CPU and GPU, and it can be observed that computational resource usage and training time are reduced.

In order to calculate the processor and memory utilization, the GFTL model was run in a controlled environment while monitoring the processor and memory utilization. For this purpose, there is a variety of tools available for different operating systems (OS), such as built-in processor and memory utilization for Windows OS and lscpu and free commands in Linux-based operating systems. For this article, the lscpu and free commands are run in separate threads that log the CPU utilization per millisecond during the experiments.

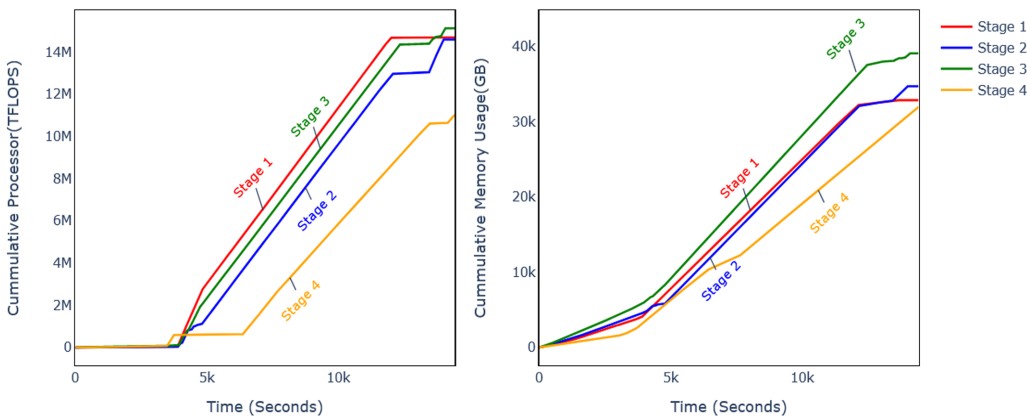

**Figure 11  Comparison of memory and CPU usage.**

**Table 5  Analysis of training and feature extraction time (CPU).**

**(a) CPU 2 Core, 4 Thread**

| DSF level | FE | | Training | | | | Total | |
|---|---|---|---|---|---|---|---|---|
| | Value | % | Processor | % | Wall | % | Total | % |
| Full | 966 | – | 2,992 | – | 2,169 | – | 3,135 | – |
| 10 | 938 | 2.90% | 3,004 | −0.40% | 2,183 | −0.65% | 3,121 | 0.45% |
| 21 | 572 | 40.79% | 1,631 | 45.49% | 1,162 | 46.43% | 1,734 | 44.69% |
| 32 | 926 | 4.14% | 2,340 | 21.79% | 1,824 | 15.91% | 2,750 | 12.28% |

**(b) Nvidia Quardpro Series**

| DSF level | FE | | Training | | | | Total | % |
|---|---|---|---|---|---|---|---|---|
| | Value | % | Processor | % | Wall | % | | |
| Full | 2,686.9 | – | 52 | – | 425 | – | 3,111.9 | – |
| 10 | 2,610.7 | 2.84% | 49.9 | 4.04% | 421 | 0.94% | 3,031.7 | 2.58% |
| 21 | 2,540.2 | 5.46% | 45.5 | 12.50% | 410 | 3.53% | 2,950.2 | 5.20% |
| 32 | 2,393.6 | 10.92% | 33.1 | 36.35% | 335 | 21.18% | 2,728.6 | 12.32% |

**(c) Analysis of offloading computational complexity (GPU)**

| DSF level | Parameters | Features extraction (seconds) | | | |
|---|---|---|---|---|---|
| | | Normal | Benign | Malignant | Total |
| Full | 23,587,712 | 13s | 17s | 16s | 56s |
| 10 Layers | 19,115,904 | 13s | 17s | 15s | 55s |
| 21 Layers | 14,644,096 | 13s | 16s | 15s | 54s |
| 33 Layers | 8,589,184 | 4s | 15s | 14s | 33s |

The memory and processor utilization are measured at different intervals, such as feature extraction, training, and validation. The CPU and GPU utilization were both measured and converted to equivalent processor excitation unit Tera Floating Point Operations per Second TFLOPS.

Figure 10 shows the cumulative processor and memory usage for the twelve minutes during which all the phases of experiments were performed consecutively. It shows that processor usage was reduced by 25% and memory usage by 22%. The processor usage is measured in equivalent TFLOPS. Table 5 shows the further analysis of different experiments' phases for CPU (Table 5a) and GPU (Table 5b), and different parameters are also shown in Table 5c.

The feature extraction time is reduced by 2.90%, 40.79%, and 4.14% with CPU and 2.84%, 5.46%, and 10.92% for GPU on DSF levels 10, 21, and 32, respectively. The wall time during training is reduced by −0.65%, 46.43%, and 15.91% on respective DSF discard levels, and processor time is reduced by −0.40%, 45.49%, and 21.79%.

### Future research directions

The main contribution of this article is to evaluate an improved transfer learning model with better accuracy and reduced complexity. The article has proposed an improved transferred learning strategy and validated it by detecting and classifying breast cancer more efficiently and accurately. For this purpose, different evaluation measures are used to check the model's performance: accuracy, recall, precision, and F1-score. The results of the above evaluation metrics show that our model satisfied all the requirements for optimal breast cancer detection and classification.

This article has evaluated the effects of discarding the DSFs on the ResNet50 model using the CBIS-DDSM dataset. However, as discussed in the analysis of the previous studies, there are other models and datasets with comparable performance. There is a need to apply the GFTL strategy to recent models for BCDC and evaluate their performance and computational requirements.

Furthermore, it is expected that further elimination of the DSFs can reduce computational requirements. Still, applying suitable techniques like fine-tuning and discrepancy distribution can improve the BCDC performance. However, further studies are required to evaluate the above aspects and their effect on performance and computational complexity.

In addition to the above, the study used all the images of the dataset, which may have some unnecessary details. Suitable feature selection techniques can potentially improve the model performance and computational resource efficiency of BCDC. However, improving the accuracy and computational resource efficiency with GFTL and feature selection on larger datasets requires further research.

## CONCLUSION

In this research study, this article has presented an improved transfer learning strategy with better accuracy and reduced complexity. The proposed methodology has been validated on the CBIS-DDSM dataset through the detection and classification of breast cancer. The main objective of the research is to study the convolution base of the pre-trained CNN model ResNet50. The impact of generic feature sets and dataset-specific features has been studied on the classification results. In the proposed methodology, different higher layers of the convolution base have been discarded, and then the features from the modified

model. The extracted features have been given to the classifier, and it was observed that the higher layers of the pre-trained model ResNet50 are not beneficial in breast cancer detection. However, the earlier layers are more beneficial as these are used to extract generic features like edges and textures. It has been shown that discarding the DSSF to different levels of the convolution base increases the accuracy by 7% and reduces the computational requirements for training time by 12%, processor utilization by 25%, and memory usage by 22%. The feature extraction time is decreased with the GFTL strategy due to a reduction in the significant number of parameters.

### Funding
The authors received no funding for this work.

### Competing Interests
The authors declare there are no competing interests.

### Author Contributions
- Muhammad Safdar Ali Khan conceived and designed the experiments, performed the experiments, analyzed the data, performed the computation work, prepared figures and/or tables, and approved the final draft.
- Arif Husen conceived and designed the experiments, performed the experiments, analyzed the data, performed the computation work, prepared figures and/or tables, authored or reviewed drafts of the article, proofreading, and approved the final draft.
- Shafaq Nisar conceived and designed the experiments, performed the experiments, performed the computation work, prepared figures and/or tables, authored or reviewed drafts of the article, proofreading, and approved the final draft.
- Hasnain Ahmed performed the experiments, analyzed the data, authored or reviewed drafts of the article, proofreading, and approved the final draft.
- Syed Shah Muhammad performed the experiments, authored or reviewed drafts of the article, and approved the final draft.
- Shabib Aftab analyzed the data, authored or reviewed drafts of the article, and approved the final draft.

### Data Availability
The code and forked dataset is available at Zenodo: Husen, A. (2024). GFTL Main Code [Data set]. Zenodo. https://doi.org/10.5281/zenodo.10568721.

The data is available at Kaggle: https://www.kaggle.com/datasets/awsaf49/cbis-ddsm-breast-cancer-image-dataset.

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
