# Peer review of "Offloading the computational complexity of transfer learning with generic features"

_PeerJ Computer Science, doi:10.7717/peerj-cs.1938_

## Round 0.1 · original submission · Major Revisions

Please revise the document, taking into consideration the feedback provided by both reviewers.

Reviewer 2 has requested that you cite specific references. You may add them if you believe they are especially relevant. However, I do not expect you to include these citations, and if you do not include them, this will not influence my decision.

**Language Note:** The review process has identified that the English language must be improved. PeerJ can provide language editing services - please contact us at copyediting@peerj.com for pricing (be sure to provide your manuscript number and title). Alternatively, you should make your own arrangements to improve the language quality and provide details in your response letter. – PeerJ Staff

Reviewer 1 ·

Basic reporting

The authors largely use clear, professional English in the manuscript to explain definitions of all relevant terms; however, the manuscript has introduction and methods sections with various redundancies and together read more like a textbook than a findings-driven paper. The manuscript could be substantially improved if redundancies were removed, the introductory narrative streamlined, more claims were supported by citations, and the methods were divorced from continued literary review. For example, please note the following major and minor suggestions:

Major:

The following lines with claims not cited and/or phrases that were redundant:
L37, L65-L66, L69, L149-L151, L159, L202, L260, L269, L290, L330, L395-401, L436-439 – common phrases included « saves time and resources” and “mammography is considered a gold standard”.

L179-181, please consider rephrasing to mention first what are ducts, and then how they are common spots for breast cancer attacks with a citation. Please also note, the breast cancer research and science in this manuscript often reads as scattered throughout the manuscript rather than constructively advancing the literature review.

L232-L244, considering offering a couple case studies and their values showing how others methods directly compare to the authors’ research, so as to give the reader a stronger reference point as to where this research lands in the literature.

L246-L255, there is a jump from breast cancer impacts (report not cited) to BCDC without a clear connection between the two concepts. Section 2.3 can also be assimilated into earlier sections as it was redundant in that its integration into earlier sections could help create one literature review instead of two forked reviews.

L294 is where I note that the authors begin to demonstrate what they have done to conduct their research. I would find here to be the appropriate place to mark the start of the methods.

L387-L417 reads as a mix of methods and literature review and could benefit from further citations.


Minor:
L18, please consider removing “The” from “The deep learning approaches”.
L30, please consider replacing “minimizing” with “minimizes”.
L32, please consider stating “transfer learning (TL)” instead of only its acronym TL.
L180, please consider not using “BC” as an acronym for “breast cancer” if it and whenever it is easier to read “breast cancer”.
L183, please decapitalize the “L” in “Lobes”.
L416 and L418, please replace the adjected “excellent” with more cited metrics or descriptions further demonstrating its capabilities.


Finally, please consider how figures and tables could also be improved to elevate its technical use and appearance by doing the following:

All figures and tables could benefit from fewer acronyms.
Table 1 could benefit from a legend to confirm what the different marks suggest.
Figures 11 and 12 could be combined into a single figure and their aesthetics could match the styling used in Figure 8.

Experimental design

The authors follow a technical, robust procedure, and the methods of the study were outlined; however, the methods were not clearly described in sufficient detail and information to replicate. The methods also bleed into the results section (e.g., L511-515). Authors could remove literature review in the methods section to determine how much information was related to the method’s justifications and experimental testing design only. This would also help a reader determine which steps help answer the purpose of the study as outlined in L447-452 (i.e., the efficacy of offloading) if one were to replicate the study.

Validity of the findings

Findings were robust and can be meaningfully compared to if replicated. Conclusions also linked to the authors’ original research question.

Reviewer 2 ·

Basic reporting

Clarify Technical Implementation: Detail the specific process of discarding domain-specific features from pre-trained models. Highlight the methods or algorithms used for feature selection or elimination.

Thorough Methodology Description: Expand on the methodology used to evaluate performance metrics like precision, recall, and F1-score. Provide a step-by-step explanation of the metrics' calculations.

Dataset Description: Elaborate on the breast cancer dataset used for experimentation. Include details regarding its size, characteristics, any preprocessing steps, and how it was curated.

Comparison with Baseline Models: Compare the proposed approach with other baseline models or techniques commonly used in breast cancer detection. This could offer a clearer perspective on the innovation's significance.

Computational Efficiency Analysis: Provide a deeper analysis of the computational efficiency improvements achieved. Detail how these reductions in training time, processor utilization, and memory usage were measured and quantified.

Statistical Significance: Discuss the statistical significance of the observed performance improvements. Include information on significance testing or confidence intervals for metrics like accuracy and computational requirements.

Visualization of Results: Incorporate visual aids, such as graphs or tables, to present the performance gains and reductions in computational resources. Visual representations can help readers quickly grasp the significance of the findings.

Experimental design

Controlled Feature Elimination: Consider implementing a controlled feature elimination process. This entails systematically discarding domain-specific features while maintaining a controlled environment, perhaps through various thresholds or iterative methods, to assess the impact on model performance. This approach can offer a more comprehensive understanding of how eliminating specific features affects outcomes.

Cross-validation and Sensitivity Analysis: Incorporate cross-validation techniques and sensitivity analysis into the experimental design. Cross-validation can validate the model's performance across different subsets of data, ensuring robustness and generalizability. Additionally, sensitivity analysis helps gauge the model's stability by systematically altering parameters or thresholds to assess their impact on performance metrics and computational efficiency. These approaches can fortify the reliability of the findings.

Validity of the findings

None

Additional comments

Please avoid citing sources that were published before to 2019. Cite current research that are really pertinent to your topic. The study also lacks sufficient citations. Another critical step is to compare the topic of the article to other relevant recent publications or works in order to widen the research's repercussions beyond the issue. Authors can use and depend on these essential works while addressing the topic of their paper and current issues.

A. Heidari, N. J. Navimipour, M. A. J. Jamali, and S. Akbarpour, "A hybrid approach for latency and battery lifetime optimization in IoT devices through offloading and CNN learning," Sustainable Computing: Informatics and Systems, vol. 39, p. 100899, 2023/09/01/ 2023, doi: https://doi.org/10.1016/j.suscom.2023.100899.
Amiri, Z., Heidari, A., Navimipour, N.J. et al. Adventures in data analysis: a systematic review of Deep Learning techniques for pattern recognition in cyber-physical-social systems. Multimed Tools Appl (2023). https://doi.org/10.1007/s11042-023-16382-x
A. Heidari, N. J. Navimipour, M. A. J. Jamali, and S. Akbarpour, "A green, secure, and deep intelligent method for dynamic IoT-edge-cloud offloading scenarios," Sustainable Computing: Informatics and Systems, vol. 38, p. 100859, 2023.
A. Heidari, M. A. J. Jamali, N. J. Navimipour and S. Akbarpour, "A QoS-Aware Technique for Computation Offloading in IoT-Edge Platforms Using a Convolutional Neural Network and Markov Decision Process," in IT Professional, vol. 25, no. 1, pp. 24-39, Jan.-Feb. 2023, doi: 10.1109/MITP.2022.3217886.

---

## Round 0.2 · Minor Revisions

Your submission will be considered for acceptance once you have addressed the major suggestions outlined in the basic reporting section by reviewer 1.

Reviewer 1 ·

Basic reporting

The authors have improved their citations and provided a stronger demonstration on where their research stands in the ML literature, especially after adding a legend in Table 1. The manuscript could still benefit from improvements, specifically where the writing lacked citations in the medical literature to back up authors’ claims on breast cancer reporting and research. Likewise, for claims on ResNet50, it would be beneficial to have more citations. For example, please note the following major and minor suggestions:

Major:
Section “1.3. A review of BCDC” lacks citations in most instances where the authors make medical claims. Examples include but not limited to: 170-176 and 185-186.
L402-403, please consider citing this quotation.
L416-418 and L424-425, please consider citing these claims.

Minor:
L46, please consider removing the word “only”.
L81-L83, please consider briefly telling how transfer learning has been shown to solve the problem of scarce data. Since the manuscript alludes to data augmentation in the methods, this could also be introduced here in passing.
L239, please consider replacing the word “ladies” with “women” and adding the report in the citations.
L467-468, please uncapitalize “Normal”, “Benign” and “Malignant”.

Experimental design

The authors improved the replicability of their study by providing further details and information on their dataset and relevant metric calculations. The authors also do address noise in the DDSM images using a median filter but don’t address concerns on any foreseen, quality limitations of DDSM. Where necessary or relevant, could the authors detail their handling, justification, and/or level of concern around any known biases that could exist in the dataset or from using ImageNet-trained CNNs? If so, this could either be introduced in the methods or addressed as potential limitations in the Evaluation or Results section.

I also noticed that sections like Section 2.6 still contain detailed background information that may be outside the purpose of a Methods section that largely provides concrete steps and justifications so that others can reproduce their work. Likewise, processor, memory utilization, and model metric calculations were outlined in the Results section rather than the Methods section. I may defer to the authors to decide where is most effective to place background details, methods reporting, and results in their work, but I suggest a more expected format for a tighter organization if possible.

Validity of the findings

With the new contributions in the results, I continue to support that the findings were robust and can be meaningfully compared to if replicated.

Reviewer 2 ·

Basic reporting

No comment.

Experimental design

No comment.

Validity of the findings

No comment.

Additional comments

No comment.

---

## Round 0.3 · accepted · Accept

I am pleased to inform you that your manuscript titled "Offloading the computational complexity of transfer learning with generic features" has been accepted for publication in PeerJ Computer Science. The reviewers were highly positive about the quality and significance of your work, and after careful consideration, we believe that your study makes a valuable contribution to the field.